# A Site-Perspective on the Second Sophistic of the near East and Its Impact on the History of Rhetoric: An Overview

Richard Leo Enos

Department of English, AddRan College of Liberal Arts, Texas Christian University, Fort Worth, TX 76129, USA; r.enos@tcu.edu

**Abstract:** This essay introduces and examines the impact of the Second Sophistic in the Near East on the history of rhetoric. Although the overall impact of sophists is apparent as early as the Classical Period of ancient Greece, this work emphasizes the renaissance of sophistic rhetoric during the so-called Second Sophistic, a movement that flourished slightly before and throughout the Roman Empire. The Second Sophistic provided an educational system that proved to be a major force spreading the study and performance of rhetoric throughout the Roman Empire. This essay examines and synthesizes scholarship that employs conventional historical approaches, particularly research that often focuses on individual sophists, in order to establish a grounding (and justification) for concentrating on what is termed here as a "site-perspective." That is, this essay stresses the importance of the sites of sophistic education and performance, arguing for such an orientation for future research. This essay also advances observations from the author's own experiences and research at ancient sites in Greece and Turkey, as well as other sources of archaeological and epigraphical research. Such work reveals that artifacts at archaeological sites—epigraphy, statuary now held at museums in Greece and Turkey, and a range of other forms of material rhetoric—provide contextual insights into the nature, influence, and longevity of rhetoric during the Second Sophistic beyond examining the achievements of individual sophists. A site-perspective approach reveals that a symbiotic relationship existed between the educational achievements of the Second Sophistic—in which rhetoric played a major role—and the social and cultural complexities of the Roman Empire. Such observations also reveal the benefits, but also the need, for further fieldwork, archival research, and the development of new methodological procedures to provide a more refined understanding of the impact of the Second Sophistic on the history of rhetoric.

**Keywords:** Greece; rhetoric; Roman Empire; Second Sophistic; sophists; Turkey

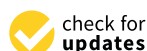

> In Greek thought as a whole, the Sophists occupy a spearhead position, one which they themselves assumed in a methodical fashion which has sometimes led to misunderstandings. They have suffered from the lack of understanding to such a degree that their influence is now sometimes hard to discern. Without them, however, those who followed, with their unflagging fervour and their sense of pathos, tragedy, and history, would never have been as they were. Nor, indeed, should we be as we are today.
>
> —Jacqueline de Romilly, *The Great Sophists in Periclean Athens* (De Romilly 1992)

## 1. Introduction: Western Sophists and the Founding of Rhetoric as a *Téchne*

Accounts of the origins of the history of rhetoric as an art (*téchne*) date back to the early decades of the fifth century BCE and credit Corax and Tisias of Syracuse as its founders. These early accounts also credit Gorgias of Leontini with introducing rhetoric to Athens when he came to represent his city as a *presbyter* (Enos 1992; Enos 2012, pp. 93–108). This early period of rhetoric is often called the "First Sophistic." Although the narratives portraying this origin of rhetoric have been contested, current research on this early period has enriched

our understanding of this nascent period while also revealing its complexities (Schiappa 1990, 1999; Cole 1991; Knudsen 2014). Most recently, Laura Viidebaum argues convincingly in her *Creating the Ancient Rhetorical Tradition* that ancient rhetoric is far from being unified either in theory, education, or practice. Her primary attention to revealing the competing, and sometimes polemical, views of Hellenic rhetoric concentrates on the distinctly different paths of Lysias and Isocrates. As Viidebaum demonstrates, Lysias's orientation to oratorical style and persuasion stands in stark contrast to Isocrates's politico-philosophical treatment of literate rhetoric (Viidebaum 2021, pp. 4, 62, 244). Both Lysias and Isocrates had a profound impact on rhetoric and make apparent the need for historians of rhetoric to recognize the different, often competing strains of rhetoric that shaped the discipline and that need to be recognized to understand the complicated forces that influenced the Second Sophistic. Such complexities require that we not only re-examine such Attic orators as Lysias and Isocrates but even such prominent thinkers as Plato when trying to understand the foundations of ancient rhetoric. For example, Edward Schiappa (1990) argues that it may have been Plato who first discussed rhetoric as a *téchne*. Regardless of when rhetoric formally began as a *téchne*, however, the seminal role and involvement of sophists such as Gorgias in this nascent period of the history of rhetoric has not been questioned, leading us to conclude that rhetoric as a *téchne* was created by sophists.

Despite the controversy over the founding of rhetoric, a few issues are indisputable. First, sophists such as Gorgias were already established and (in his case) admired during Greece's Classical Period. As Scott Consigny (2001) argues in *Gorgias: Sophist and Artist*, Gorgias was a respected thinker who should be recognized for offering impressive rival "philosophies" to those of Plato and Aristotle. Second, the Greek population of the island of Sicily and Southern Italy (i.e., Magna Graecia) was the result of colonization by several city-states of Eastern Greece, who brought to the West not only citizens but also their rhetorical culture. Third, we know that the later movement called the "Second Sophistic" (a term coined by Philostratus in the third century CE) thrived throughout the Roman Empire and had its own origin in the Greek East and the Ionian coast of Anatolia that is now part of modern Turkey (see Figure 1). In his seminal work, *Greek Sophists in the Roman Empire*, G. W. Bowersock claims that "the origins of the sophists are to be found all over the Greek East" and lists several cities that are associated with sophists (Bowersock 1969, p. 21).

Seminal research on the sophists has contributed to an understanding of their individual thought and practice. However, my own research at archaeological sites in Greece, as well as my experiences in Turkey, have led me to the belief that much more can be learned by focusing on sites of the study, practice, and performance of rhetoric in the Greek East. Attention to evidence provided by extant artifacts—the material rhetoric found at archaeological sites, local museums, and regional depositories (*apotheki*)—furnishes important information about the context or *kairos* of the Second Sophistic in the Roman Empire.[1] The social complexities and power dynamics of the Roman Empire that affected the sophists of the Second Sophistic have been made clearly evident by scholars such as Susan C. Jarratt (e.g., *Chain of Gold* (Jarratt 2019)). Jarratt's work is especially important for the thesis of this work. Although Jarratt's recent work, *Chain of Gold*, concentrates on the rhetorical analysis of what she terms "second sophists", she does so by making apparent the social complexities and power relationships within the Roman Empire—including cultural and even geographical issues—that sophists had to face. Because of such complexities, focusing on the social climate of these times helps to reveal that the broad-based and recursive impact of the Roman Empire nonetheless facilitated, and in return was nurtured by, the Second Sophistic. The symbiotic relationship that existed between Roman patronage of Greek sophists of the Second Sophistic and the educational benefits derived from that patronage not only served educational purposes but also proved to be a unifying social and cultural force throughout the Empire. Understanding the rhetorical impact of the Second Sophistic from this view requires a change of perspective from West to East and from individual sophists to geographical sites of sophistic teaching and performance.

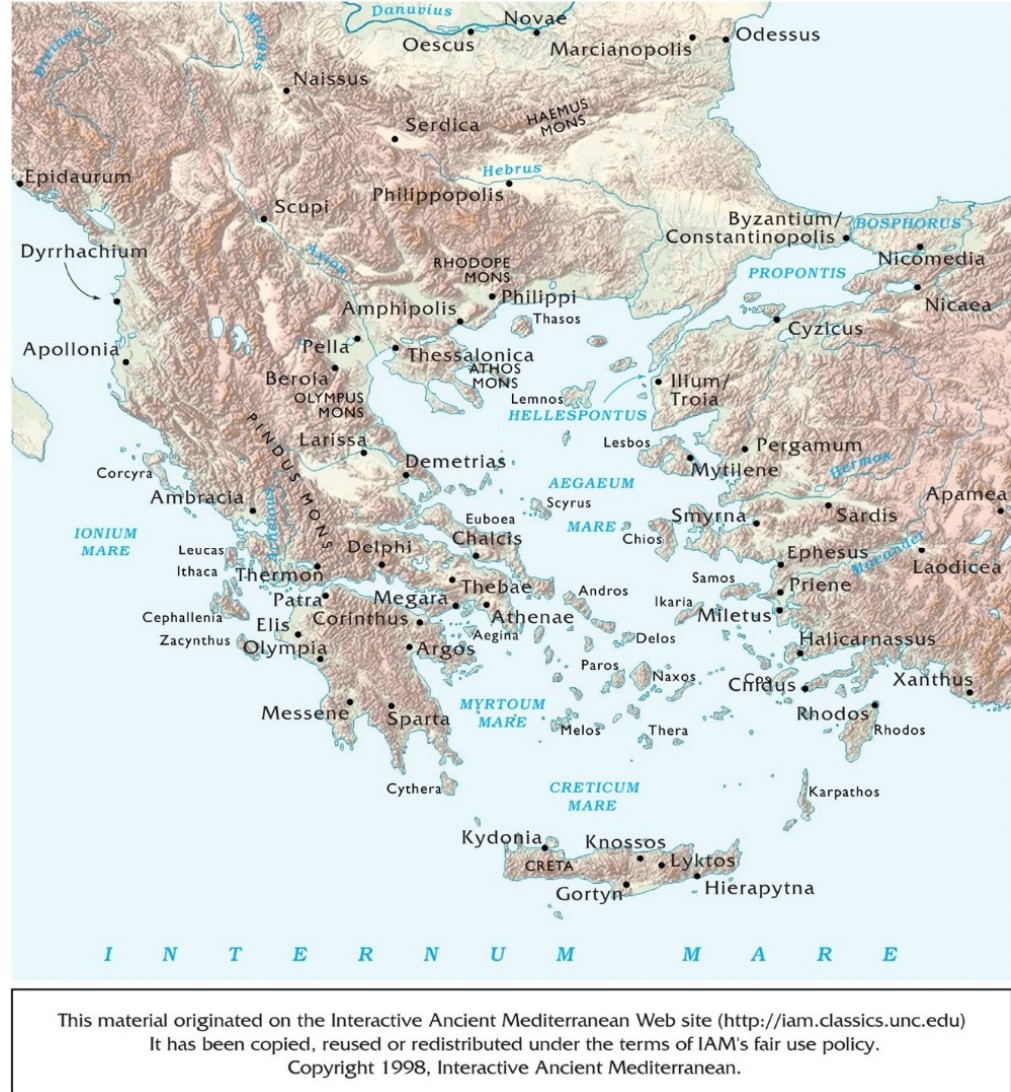

**Figure 1.** Aegean Sea and Greece. Ancient World Mapping Center © 2023 (awmc.unc.edu; accessed on 17 May 2022). Used with permission.

Another perspective that this body of scholarship argues should be altered is the identification of ancient Greek rhetoric with Athenian rhetoric. In fact, as Eric A. Havelock notes, our fixation on equating Athens and Greek rhetoric—what Havelock calls "Atheno-centrism" in *The Muse Learns to Write* (Havelock 1986, p. 125)—has encouraged us to ignore the rest of Greece, other manifestations of rhetoric, and other (albeit later) sites that were centers for the study and practice of rhetoric in the East. Of course, the attempt to recapture the early history of Greece and Anatolia in general has been a longstanding enterprise dating back at least to the seventeenth century. From our present-day view, we may be inclined to see many of these modern attempts to understand rhetoric in Greece and the Near East as motivated by imperialism and tainted by assumptions about the superiority "European" culture, but there were nonetheless efforts by Westerners to rediscover the lost histories of the ancient Hellenic world.[2] That noted, efforts by historians of rhetoric to undertake the same sort of research—albeit motivated by a more cross-cultural perspective—in the eastern parts of Greece and the Ionian coast are only at a nascent stage. This essay therefore seeks to provide an overview that not only sheds new light on the Second Sophistic directly but also indirectly serves as an inducement promoting more fieldwork and archival research by contemporary rhetoricians that will contribute to a more accurate and culturally sensitive history that has been lost over the centuries (see Figure 2).

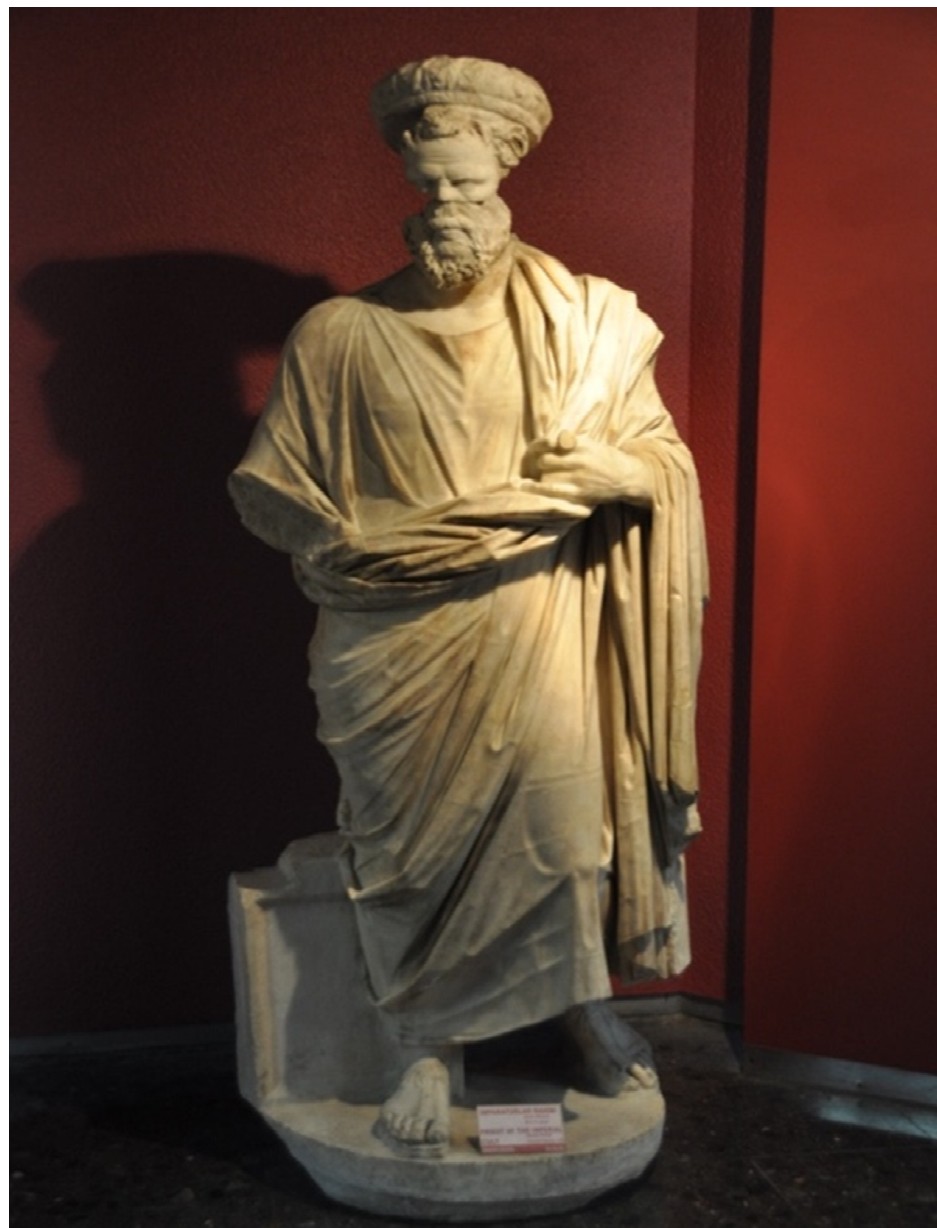

**Figure 2.** Ephesus, priest of the Imperial Cult. Also identified as a portrait statue of Flavius Damianus, the sophist as imperial priest (see Akurgal, Plate #52). Izmir Archaeological Museum. © Livius.org. Jona Lendering. Used with permission.

## 2. Greek Sophistic of the near East: A Brief Overview of Sources and Scholarship

In order to realize and appreciate the benefits of the site-perspective that is argued for in this essay, it is necessary to synthesize and evaluate earlier foundational sources and research. The primary works of Philostratus (*Vitae sophistarum*) and Eunapius (*Vitae sophistarum*) are invaluable primary sources for the study of Greek sophists of the Near East. Likewise, the nineteenth-century three-volume edition of Leonard Spengel's *Rhetores Graeci* (Spengel 1853–1856) and Konradus Jander's *Oratorum et Rhetorum Graecorum Fragmenta nuper Reperta* (Jander 1913) offer an array of primary texts by rhetoricians from both the Classical Period and the Second Sophistic. George Kennedy's *Progymnasmata: Greek Textbooks of Prose Composition* (Kennedy 1999) is an excellent companion to the primary sources mentioned above. Those looking for a survey that introduces specific sophists will find Donald C. Bryant's (et al.) *Ancient Greek and Roman Rhetoricians: A Bibliographical Dictionary* (Bryant 1968) and Project Rhetor Director James J. Murphy's *A Bibliographical*

*Dictionary of Greek and Roman Rhetoricians: A Preliminary Survey* (Project Rhetor 1985) very helpful synoptic overviews of sophists in terms of time, location, and rhetorical emphasis. Examples of in-depth studies of prominent individuals of the First Sophistic can be found in Edward Schiappa's *Protagoras and Logos: A Study in Greek Philosophy and Rhetoric* (Schiappa 1991), Scott Consigny's *Gorgias: Sophist and Artist* (Consigny 2001), and Susan C. Jarratt's rhetorical analysis of orations of prominent sophists of the Second Sophistic in *Chain of Gold* (Jarratt 2019). Those who wish to examine a detailed study of a prominent sophist of the Second Sophistic should refer to Raffaella Cribiore's *Libanius the Sophist: Rhetoric, Reality and Religion in the Fourth Century* (Cribiore 2013).

Historical scholarship provides a wealth of introductory material and important research contributions essential for continued work on sophistic teaching and practice in the Near East. G. W. Bowersock's *Greek Sophists in the Roman Empire* (Bowersock 1969), George Kustas's *Studies in Byzantine Rhetoric* (Kustas 1973), and George A. Kennedy's *Greek Rhetoric under Christian Emperors* (Kennedy 1983) remain stellar foundational works for the study of Greek sophists of the Second Sophistic and Byzantine Empire. Those who wish to consult works that offer a broad review of education in the Roman Empire will find H. I. Marrou's *Education in Antiquity* (Marrou 1985), M. L. Clarke's *Higher Education in the Ancient World* (Clarke 1971), and Richard Leo Enos's *Roman Rhetoric: Revolution and the Greek Influence* (Enos 2008) helpful treatments of the place of rhetoric in the Roman Empire.

Readers who take up the recommendation to approach the Second Sophistic from a "site-perspective" point of view should consult *The Princeton Encyclopedia of Classical Sites* (1976) for a review of archaeological discoveries at ancient sites. Detailed studies of rhetoric at specific sites are available in Robert W. Smith's *The Art of Rhetoric at Alexandria* (Smith 1974) and Raffaella Cribiore's *The School of Libanius in Late Antique Antioch* (Cribiore 2007).[3] Shorter, article-length studies of rhetoric at eastern sites include Richard Leo Enos's essays on rhetoric at Rhodes (Enos 2004), Halicarnassus (Enos 2021b), and the Dorian Hexapolis (Enos 2022). Cemil Toksöz's *Ephesus: Legends and Facts* (Toksöz 1976) provides a clear and direct overview of the Roman capital of Asia Minor and the buildings that helped to make it a major intellectual center attracting sophists. Examples of detailed historical studies of lesser-known sites are Mustafa Adak and Peter Thonemann's *Teos and Abdera: Two Cities in Peace and War* (Adak and Thonemann 2022) and Paul Cartledge's *Thebes: The Forgotten City of Ancient Greece* (Cartledge 2020). Of course, the few works mentioned here are intended as an introduction to the study of sophists in the Greek East and must be supplemented by more recent scholarship. These works, as mentioned above, provide a foundation for the study of specific subjects. Full citations for these and other works are listed in the "References" section of this essay.

## 3. Who "Counts" as a Sophist?

The term "*sophos*" has a history that is long and complex. As Christopher Moore has revealed in his recent book, *Calling Philosophers Names*, from the Pre-Classical Period and through the Classical Period *sophos* and *philosophos* have an intertwined and complicated etymological history, beginning with the Seven Sages. By the sixth century BCE, however, *sophoi* came to be understood as intellectuals who functioned as "wide-ranging civic and domestic advisers" (Moore 2020, p. 6). Throughout antiquity, as Moore points out, this pragmatic orientation also characterized many *philosophoi*, to the extent that Cicero recognized the value of philosophy when united with rhetoric (Moore 2020, pp. 67–68). Edward Schiappa also has provided us with a thorough and insightful analysis of the possible origins of the term "*sophos*" and of the implications of that collective term for such thinkers as Protagoras (Schiappa 1991, pp. 3–19). In the same spirit as Schiappa's analysis, one question we should ask ourselves in terms of the Second Sophistic is, "Who 'counts' as a sophist?" Some individuals willingly described themselves as sophists. However, individual thinkers such as Protagoras, Schiappa argues, should not be grouped with sophists and warns readers of the problems of having a collective meaning for "*sophos*" (see Schiappa 1999, pp. 48–65). Some considered "sophist" a term of derision. For example,

in the Classical Period Aristophanes chided Socrates in his play *Clouds* for being no different than any other sophist in his treatment of rhetoric. In the Second Sophistic, Lucian—a rhetorician for much of his career—employs his biting wit to mock "pseudo-sophists" who are little more than pompous blowhards or *soloikízonta* ("The Pseudo-Sophist or Solecist") and those "so-called" schools of rhetoric that intentionally avoid the rigors of classical rhetoric and merely teach histrionic, clap-trap techniques ("The Professor of Rhetoric" or *Rhetorum praeceptor*). It is easy to see, even from these few examples, that the term "sophist" can be both an expression of esteem as well as a label carrying a social stigma. We should also consider whether the term "sophist" not only changes over time but also in context; that is, we should attend to variations in the meaning and value of the term "sophist" across different regions of the Hellenic and Roman worlds. With all of these caveats noted, it is better to maintain a wide latitude of meaning for the term "sophist" rather than seek a narrow meaning that, by default, excludes individuals who in all other ways should be regarded as sophists. As will be discussed in more detail in what follows, we should also expand our research from individual sophists and look to "movement" studies and consider whether the Second Sophistic is better understood as a social movement. That is, we should ask whether the Second Sophistic is analogous to what modern researchers of persuasion and social movements consider to be a movement that is rhetorical in its features. Such a macroscopic approach will enable us to better determine whether—and despite the lack of a coherent, unifying force—there was a widespread sophistic movement in the history of rhetoric and, if so, the nature and consequences of such a movement and its collective understanding of the term "sophist".

These qualifications noted, we do know that the term "sophist" was used widely in ancient Greece with some shared meaning. That is, and in its most basic manifestation, we should think of the term "sophist" the way we think of the Latin term "*senex*", that is, as a noun that also has the qualities of a gerund. In the case of *senex* the meaning would come from the condition (or even activity) of being or growing old. In Greek, the term "*sophos*" would fulfill the same act–essence association—that is, one who says or does things that society considers to be wise so that such activities or statements make the person wise in essence or being. Applying a reception-theory approach grounds the meaning of critical concepts such as "sophist" in the time and place of its use and not in the view of the researcher.[4] John Poulakos, for example, applies reception theory in *Sophistical Rhetoric in Classical Greece* (Poulakos 1995) to provide a better understanding of how sophists of the Classical Period were viewed and how they were "received" by Plato, Aristotle, and Isocrates. We, in turn, should do the same with respect to how the term "sophist" was received during the Second Sophistic. We also can study the meaning of the term "sophist" in context through an examination of material evidence. Epigraphical sources, for example, illuminate the use of the term "*sophos*" in a variety of ways, such as the recording of victors of rhetorical and literary contests, including contests of sophistic declamation such as those held at the Amphiareion of Oropos (*IG* 1892, 7.1 no. 414, line 8, ΣΟΦΙΣΤΗΣ). Finally, we should realize that not all sophists had rhetoric as a central concern. We can claim, as Laurent Pernot astutely does, that all sophists used language to share their wisdom and that their use of language therefore qualifies them for attention by rhetoricians. As Pernot also points out, many sophists offered theories of rhetoric (Pernot 2005, pp. 189–90; 2017, pp. 255–56). In sum, it is best, as we gather insights across time, across Greece, and throughout the Roman Empire, to be inclusive and consider any individuals who fall into the categories mentioned above as sophists worthy of our study by how they were received in their context and at their site(s) of teaching, practice, and performance.

## 4. Sophistic Rhetoric: Shared Features

In order to better understand the Second Sophistic, we should begin by understanding the features of the First Sophistic of the Classical Period. Although "classical rhetoric" appears to be a univocal term, there are actually strains, and even sub-strains, of ancient Greek rhetoric (Murphy 1974, pp. 3–42; Viidebaum 2021, pp. 17–135). Platonic and Aris-

totelian rhetoric centers on the philosophical and theoretical perspectives of rhetoric as a discipline. The Platonic approach investigates, through the dialogues and their principal character, Socrates, the foundational nature of rhetoric; what, if anything, rhetoric contributes in the quest for knowledge; and whether rhetoric warrants being characterized as *paideia*—that is, an art or *téchne* whose objective is the virtuous quest for intellectual excellence (Jaeger 1943–1945). Aristotelian rhetoric views rhetoric as a *téchne* that facilitates the discovery and expression of arguments whose merits and validity are adjudicated by auditors (Grimaldi 1990). Isocratean rhetoric embraces both orality and literacy and is oriented toward a more global, social perspective than either of the characterizations of rhetoric by Plato or Aristotle. In *Education in Antiquity* Marrou claims that the contributions of Plato and Aristotle constitute the foundational pillars upon which the tenets of classical education were built, but that "in the eyes of posterity, it was Isocrates who carried the day" (1985, p. 224; cf. Walker 2011, p. 6 *et passim*). "Isocrates believes", de Romilly argues, "that it is human opinions that provide the authority in matters of morality" (De Romilly 1992, p. 211). It is the contention of this essay that of Plato, Aristotle, and Isocrates, the work of Isocrates is most in harmony with the sophists, to the extent that another pillar to the foundation of classical rhetoric must be acknowledged: the contributions of the sophists, particularly in rhetoric and especially in Eastern Greece and Asia Minor. In fact, in the Platonic, Aristotelian, and Isocratean orientations of rhetoric, the accounting is often done in terms of comparing and contrasting the practices of contemporary sophists.

What, then, are the features of sophistic rhetoric? To begin with, sophistic rhetoric is grounded in pragmatism, expedience, and social relevance. There is no doubt that the accolades from scholars on the contributions to rhetoric by Plato, Aristotle, and (more recently) Isocrates are well deserved. That said, rhetoricians such as Susan Jarratt, especially in her *Rereading the Sophists: Classical Rhetoric Refigured* (Jarratt 1991), make a convincing case that equating Aristotelian rhetoric with classical rhetoric, as well as assuming the inherent superiority of Aristotelian rhetoric over sophistic rhetoric, are views that need to be reconsidered because they do a disservice to the rival paradigm of rhetoric offered by sophists. The long-term, widespread, and socially relevant heuristics of sophistic rhetoric had perhaps the greatest impact in the history of rhetoric, if determined by the range and degree of influence. In fact, the value and benefits of sophistic rhetoric in the Classical Period were so obvious to sophists of the Roman Empire that they promoted a renaissance of Greek rhetoric.

A second feature of sophistic rhetoric Is that it is often associated with the ornate Asiatic style popularized by the virtuosic epideictic speeches of Gorgias, which is contrasted with the plain Attic style and the middle Rhodian style. The Asiatic style was subjected to severe criticism by Atticists, who characterized the Asiatic style of eastern sophists as bombastic, histrionic, and inappropriate for such venues as the lawcourts. As Quintilian notes, some attribute these differences to the belief that as Greek cities spread into Asia there was a desire for eastern Greek rhetoricians to display mastery of rhetoric and to do so by ornamentation and what mainland Greeks (such as Athenians) saw as excessive theatricality. Quintilian, however, disagrees and views such differences as driven by the cultural preferences of orators and listeners (Quintilian, *Institutio oratoria* 12.10.16–19). In this passage, Quintilian also points out that these eastern Greeks also developed the Rhodian style, which was viewed as a stylistic register somewhere between Attic and Asiatic rhetoric.

A third feature of sophistic rhetoric is its concern with the teaching of composition, argument, and performance. In *The Great Sophists in Periclean Athens* de Romilly makes many penetrating observations about the sophists of the Classical Period, and some of these features can be carried over to the sophists of the Second Sophistic. De Romilly argues that it is not possible to understand accurately Hellenic contributions without recognizing the important part that sophists played in most areas of Greek culture (De Romilly 1992, ix, p. 234).[5] In a similar respect, the impact of rhetoric on education in the Roman Empire cannot be fully understood without understanding the sophists of the Roman Imperial Period,

particularly in the East. Sophists taught practical education, and their part in contributing to the enduring impact of rhetorical education shaped higher education itself (1992, p. 33). Despite the practical orientation of sophistic rhetoric, the sophists of the Classical Period were also famous for their "dazzling intellectual performances" that were often viewed "as the most fascinating of spectacles" (1992, p. 36). In the same respect, the performative aspect of sophists of the Second Sophistic (as alluded to above) was also popular, so much so that contests for sophistic declamation were formalized with epigraphical lists of victors at sites such as the Amphiareion of Oropos (Enos 2008; Walker 2011, p. 6). As de Romilly notes, classical sophists were often chided by prominent intellectuals such as Plato and Aristophanes, who devalued sophistic expertise as casuistry and relegated sophistic rhetoric to quibbling and eristics (De Romilly 1992, p. 55). However, as Pernot notes, declamation was valued and taught by the sophists of the Second Sophistic (Pernot 2017, p. 255). Yet, despite criticism for excessive showmanship, what did become clear in the Second Sophistic is that declamation in both Rome and Greece produced an education that stressed cogent argumentation and that the benefits of casuistry were apparent and recognized even by Quintilian (Enos 2021a).

### 5. The Second Sophistic as a Rhetorical Renaissance

Many of the observations that scholars such as de Romilly and Schiappa make about the achievements and contributions of the sophists of the Classical Period can also be said of the sophists of the Second Sophistic, who saw as their standard and model the sophists of the Classical Period and sought consciously to make the Second Sophistic a movement as well as a rhetorical renaissance (see Pernot 2017, pp. 253–54; Walker 2011, pp. 194, 212).[6] Bowersock argues that the Greek "literary renaissance of the second century owed much to long-term Roman support" and that "this intercourse between Greeks and Romans not only affected the course of Greek literature and rhetoric; it unified East and West" (Bowersock 1965, pp. 123, 146–47). A symbiotic relationship existed between the educational achievements of the Second Sophistic—in which rhetoric played no small role—and the sustained coherence of the Roman Empire. The Second Sophistic indirectly but emphatically nurtured a Graeco-Roman bond that promoted the cultural unity of the Empire. As Bowersock acutely observes, Augustus, with many other Roman emperors to follow, recognized and embraced the Roman fascination with Hellenic life (Bowersock 1965, p. 94). "It is clear", Bowersock argues, "that Augustus was well aware of the value of intelligent and loyal Greeks both in his court at Rome and in the Greek-speaking portions of the empire" (1965, p. 41). Emperor Nero, for example, was an ardent philhellene, to such a degree that he went to Greece to participate in artistic performances and even granted Greece special tax immunities and privileges because of the many rhetorical, theatrical, and literary contributions made in the Greek East (Enos 2013). During my site work at Thebes I came across an epigraphical record of a speech given by the Emperor Nero clearly revealing his ardent support and patronage of Greek culture (see Figure 3). To realize the benefit of this recursive relationship, Roman emperors actively colonized in the Greek East and, in doing so, brought Roman culture to the far reaches of their Empire, such as Alexandria and Antioch (Bowersock 1965, p. 65). Romans, in turn, were exposed to Greek culture, with such prominent emperors as Tiberius, Vespasian, Hadrian, and Nero promoting rhetoric through special privileges to sophists and building programs for educational enrichment (see, e.g., Enos 2008, pp. 138–51).

Just as we think of the Renaissance (*Il Rinascimento*) of Florence not only as a literary movement but also as a cultural movement, so also should we think of the Second Sophistic as more than a rebirth of the sophistic rhetoric of the Classical Period of Greece. The Second Sophistic can also be viewed as a social/cultural movement precisely because it was rhetorical; that is, the Second Sophistic was "rhetorical" not only in what it practiced and taught but because it promoted indirectly (but nonetheless epideictically) values that affected the Roman Empire for centuries (cf. Pernot 2017, pp. 253–65).[7] Their shared "rhetorical" features are that they were not only large in scope but also used persuasion to advocate a set of shared values that shaped culture and societal norms. Viewing the Second Sophistic

as a social movement that was rhetorical at its core provides historians of rhetoric with a set of heuristics that enable us to not only understand the impact of individual sophists but also to gain a macroscopic perspective that views sophistic rhetoric diachronically and cross-culturally. Finally, we also know that the ancient belief that mainland Greek cities expanded to the East and colonized as they did in the West is not completely accurate, for there is Bronze Age archaeological evidence of Greek habitation in the eastern sections of Greece and the Ionian Coast of Turkey. In sum, the character of sophistic rhetoric should prompt us to re-examine the contextual history within which it was taught and performed.

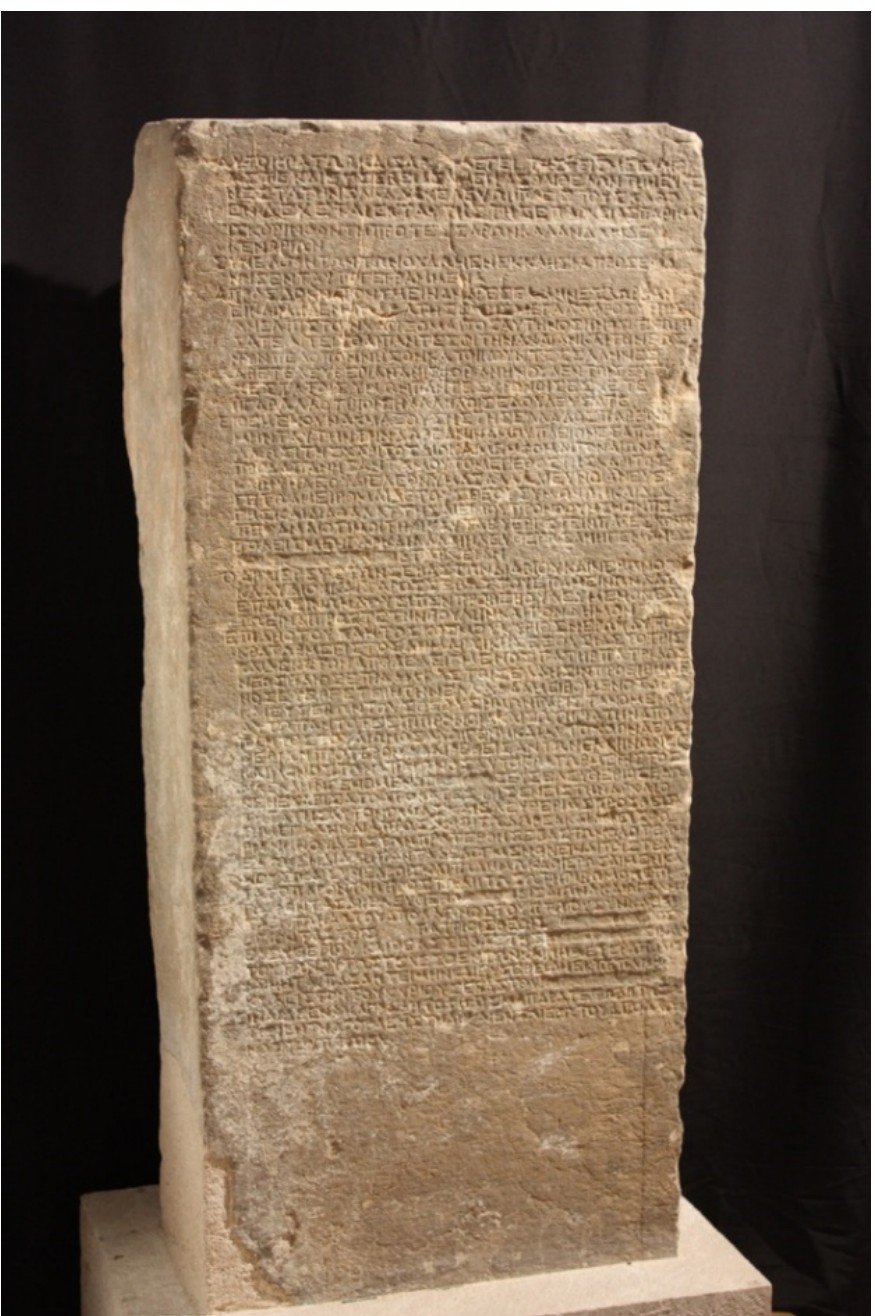

**Figure 3.** A record of Nero's speech at the Isthmian Games. Ministry of Culture and Sports. Archaeological Resources Fund. Archaeological Museum of Thebes. Used with permission.

## 6. Why Search for Sophistic Sites?

Much of what we know about the sophists comes, as discussed above, from the hostile testimony of rivals such as Plato, Aristotle, and Isocrates. In essence, the study of ancient

sophists has emphasized and examined individuals more than the sites of their origins, performance, and educational practice. In fact, those sophists who traveled to Athens to teach rhetoric were viewed as *metics*, or "non-citizen" educators, a perspective that reveals that Athenians judged such sophists as visitors or alien residents who often enjoyed some privileges but were not viewed with the same status as those holding Athenian citizenship. We have, in effect, come to know these sophists by how well they were received in prominent centers for the study and practice of rhetoric such as Athens and Rome. To be sure, acontextual research focusing on the sophists themselves has provided a great deal of insight into their talents, practices, and theories, but such research says little about the cultural climate and context of sophistic practice.

Through the research of rhetoric scholars such as Lloyd F. Bitzer (1968) and James L. Kinneavy (1986), however, we have come to realize that there is much more to be learned by understanding not only the rhetorical situations of the sophists but also the more general *kairos*—understood, again, not as the timely or propitious moment but as a social and mental climate—of the times and places within which sophistic theory and practice developed. For example, Athenaeus notes in *Deipnosophistae* (1.3) that many of the sites of sophistic rhetoric in the Greek East, Anatolia, and Egypt—such as Rhodes, Samos, Athens, Cyprus, Pergamon, and Alexandria—were also academic centers that held impressive libraries. There are even claims that Pergamon, one of the most impressive cultural sites of the Hellenistic period, held over 200,000 volumes (Akurgal 1983, p. 73). The Library of Celsus at Ephesus, famed for its beautiful façade, is believed to be the third largest library of the ancient world (following only Alexandria and Pergamon) and thought to have held over 12,000 works. As magnificent and alluring as these sites are, one wonders how many more lesser-known sites of sophistic rhetoric are awaiting rediscovery in the Greek East.

Even a cursory review of Philostratus's *Vitae sophistarum* reveals that sophists came from several locations in the Greek East. Philostratus also makes it apparent that these sophists were attracted to centers in the East, the most famous of which is Ephesus, the Roman capital of Asia Minor. Another example of a sophistic center in the East is Smyrna (modern-day Izmir), a city that not only produced many sophists but was also a center for rhetorical study and performance (cf. Jarratt 2019, pp. 42, 58). As will be discussed in what follows, the writings of Philostratus are complemented by material evidence in the Archaeological Museum of Izmir and field evidence at archaeological sites. The importance of engaging in fieldwork and on-site archival research is critical if we are to discover material evidence at such sites: we must examine the places from which, and in which, the sophists learned, taught, and performed. Archaeological research over the last century has provided a wealth of material evidence yielding a more complete and refined picture of sophistic rhetoric in Eastern Greece and Anatolia.

In sum, the treatises of Philostratus, Eunapius, Suidas, Lucian, and others have provided invaluable insight into individual sophists during the Classical Period and the Second Sophistic. However, directly experiencing the sites will be an immeasurable help in understanding the context and climate of the Second Sophistic, enabling us to better understand its social and cultural impact in the Greek East and the Ionian Coast of the Roman Empire. Some of the lost centers for the study and practice of rhetoric in this region have recently been under study, including Rhodes, Halicarnassus, Teos, Knidos, and other locations in Asia Minor (see Enos 2004, 2021b, 2022; Bean 1976; Enos and Peterman 2014; and respective entries in References). Our knowledge of these centers for the study and performance of rhetoric will provide more insight not only into other previously undiscovered centers but also into the role that these sites played in the history of the discipline. For example, through the cooperation of Turkish archaeologists, my co-author Terry Shannon Peterman and I were able to publish an inscription from Teos that reveals that a co-educational curriculum offered young women advanced studies in writing (Enos and Peterman 2014; Marrou 1985, p. 222). Even at this nascent stage of our research into material rhetoric we have learned that the paths of the sophists in the colonization of the West by prominent Greek city-states has helped to explain how rhetoric was transferred and introduced into

Magna Graecia and eventually into Rome (see Enos 2008, pp. 23–37). We also know that the renaissance of the Second Sophistic not only valorized ancient sophists of the Classical Period but also energized the study of rhetoric throughout the Roman Empire. Such findings not only enrich our knowledge of the history of rhetoric but also serve as paradigms for the potential that such work has to enrich the knowledge of rhetoric as a field. For these and other reasons, to continue to study the Second Sophistic from a site-perspective makes the potential significance of future findings worthy of our research efforts.

### 7. A Site-Perspective of the Amphiareion of Oropos: An Archaeological Approach to the Second Sophistic

The general methodological procedures for a site-perspective approach are presented in *The Sage Handbook of Rhetorical Studies* chapter "Rhetorical Archaeology: Established Resources, Methodological Tools, and Basic Research Methods" (Enos 2009, pp. 35–52). For the purposes of illustrating the benefits of this approach to examining Second Sophistic sites, this section provides a synoptic overview, drawn from a more detailed treatment in Enos (2008, pp. 152–63), of on-site work at the Amphiareion of Oropos.

The extensive and impressive archaeological artifacts in Greece and Turkey are testimonials of the Roman patronage and contributions to the Second Sophistic. Romans supported rhetoric in the Second Sophistic in other ways, however, such as sponsorship of literary and rhetorical contests at sites throughout Greece and Turkey. One site-examination of archaeological and epigraphical evidence at a small sanctuary, the Amphiareion of Oropos, reveals how examining specific sources reveals the activity and nature of rhetorical education and performance in the Second Sophistic. That is, fieldwork, archaeological reports of excavations, and in situ inscriptions enable us to reconstruct the nature and duration of rhetorical activities during the Second Sophistic. What site-perspective research, particularly in this case, makes immediately apparent is that even this small site, virtually unmentioned by ancient authors, nonetheless provides archaeological and epigraphical evidence revealing that for centuries this small sanctuary was a frequent site of rhetorical and literary contests as well as a repository of written communication about these events. Even a cursory reading of Kennedy's research makes apparent that rhetoric was taught and practiced throughout the Greek East (Kennedy 1963, pp. 3, 26, 27 *ff.*; Kennedy 1983, *ff*). On-site epigraphical sources offer evidence that is recorded contemporaneously with events and normally remains unaltered since its original composition.

For centuries the artifacts of the Amphiareion lay buried in the remote and rustic valley of the sanctuary. However, excavations that began in 1884 by the Greek Archaeological Society began to unearth material evidence of the activities of sophists of the Second Sophistic. These on-site inscriptions revealed that rhetorical and literary games honoring Amphiaraus were held from at least the fifth century BCE onward and were eventually sponsored by patrons of the Roman Empire. As is the case with most sanctuaries, the Amphiareion became an active civic center. The sanctuary was not an intellectual center such as Athens, but rather a center for display and performance; in fact, the remains of a small theatre are preserved at the site. Initially regarded as a center of healing—doubtlessly because of its excellent climate, water, and locale—the sanctuary became a resort of sorts with the games, contests, and general entertainment becoming an attraction to both Greeks and Romans. In fact, an inscription from Amphiareion reveals that Cicero was so taken by the Amphiareion that he was able to pass a decree giving the site immunities from Roman taxation (Enos 2005, pp. 457–65). Cicero's support of the Amphiareion was not unprecedented. On-site inscriptions also reveal that since at least 156 BCE the games were held under Roman auspices and were even renamed the "Amphiareion and Roman" games.

The extant inscriptions of the Amphiareion provide invaluable material evidence about the role and scope of Roman patronage and the nature of rhetorical practices during the Second Sophistic in the Greek East. In addition to documenting the practice of rhetorical contests, we also learn about individual victors and their respective home cities. Inscriptions also help to explain the role of the *agonothetos*, who had the responsibility of organizing

the games, as well as the men and boys (*epheboi*) who participated in various events. One inscription, for example, indicates that an Athenian named Pausimaxos was the victor in sophistic declamations. Other inscriptions from the Amphiareion that are now housed at the Athens Epigraphical Museum (*IG*, nos. 415, 417; *EM* 11969) list a range of rhetorical events: the composed encomium, rhapsodic exercises, and rhetorical exercises in the delivery of (usually) dramatic literature by an interpreter-actor or *hypokrites.* There is even the listing of a declamation celebrating Roman conquests.

The results of this site-perspective at the Amphiareion of Oropos reveal that rhetorical events had a place in the sacred games of the Amphiareion. Roman patronage and support appears to have encouraged declamatory and epideictic rhetoric, for the latest inscriptions emphasize rhetorical displays as popular artistic events. The nature of such rhetorical displays and sophistic practices would be unknown to modern historians of rhetoric were it not for the extant artifacts. Of course, there is still more to be studied at the Amphiareion (and other sites), but this summary of work is presented here to provide just one illustration of the contributions that a site-perspective on the Second Sophistic can yield and to encourage field and archival work at larger centers for the study and performance of rhetoric in the Greek East and the Ionian Coast of Turkey.

## 8. Examples (and Samples) of Environs of the Second Sophistic in the Greek East

Sophists of the Second Sophistic are evident throughout the Roman Empire (Bowersock 1969, p. 1; Pernot 2017). Although this essay concentrates on Eastern Greece and the Ionian Coast of modern Turkey, the activity of sophists is evident in such areas of the Empire as Spain, Gaul, Africa (Carthage), Sicily, and Magna Graecia (Southern Italy).[8] An examination of sophists reveals sites of the Second Sophist in the East. Some prominent examples of centers associated with sophistic activity include Athens (Lysias, Lucian), Rhodes (Molon), Byzantium/Constantinople (Aristophanes, Chrestus), Ephesus (Damianus), Smyrna (Euodianus, Nicetes Sacerdas), and Antioch (Libanius, Aphthonius). Some of the sophists initially came to these large centers as ambassadors (*presbyters*) representing their home cities but then remained at these sites and founded schools of rhetoric. In the Classical Period, for example, Gorgias of Leontini went to Athens as a *presbyter* and remained in Athens to teach rhetoric. During the Second Sophistic, Lucian came from Asia Minor and studied rhetoric at smaller schools, but eventually went to Athens (Enos 2022, pp. 298–99). There were also sites that did not attain the status of the major centers but still produced prominent sophists and literary figures, such as Halicarnassus (Dionysius of Halicarnassus), Teos (Nausiphanes), Caria (Menippies of Stratonicea), Lampsacus (Anaximenes), Apollonia (Apollodorus), Hierapolis (Antipater, Hierius, Demetrius), and Pergamon (Apollodorus, Aristocles).

Surveying sophistic sites from the East reveals several important features of the Second Sophistic. Many sophists taught not only rhetoric but also philosophy, with Stoicism appearing frequently as their orientation. Some Greek sophists, such as Dionysius of Halicarnassus, taught Latin rhetoric in such cities in the West as Rome, whereas other sophists taught Latin in such Greek centers of rhetoric as Antioch. There are also sophists—as illustrated by Dionysius of Thrax, who taught at Rhodes—who taught grammar along with rhetoric. Sophists were often attracted to major centers of rhetoric such as Athens because they offered municipal and imperial chairs of rhetoric (Avotins 1975). In addition, as Thomas M. Conley points out, Roman emperors Hadrian and Marcus Aurelius endowed chairs of rhetoric in eastern provincial cities such as Antioch, Gaza, Alexandria, and Ephesus (Conley 1990, p. 60). Some sophists accompanied individuals such as Cicero to sites of rhetoric in Asia Minor (Cicero, *Brutus* 314–316). Over the centuries we continue to hear of individual rhetoricians coming from, or teaching at, such cities and regions as Phaselius (Asia Minor), Tyre, Gaza, Ephesus, Abdera, Halicarnassus, Smyrna, Sardis, Byblus, Byzantium, Cappadocia, and Syria (Arabic Petraea).

By knowing even a slight amount of information about these sophistic sites and environs, we can make some important observations.[9] First, those sophists who were credited

with founding the dominant style of the Second Sophistic, and even the Second Sophistic itself, such as Hegesias of Magnesia and Niceles Sacerdos of Smyrna, represent the basic tenets of Asiatic rhetoric. As has been observed, the goal of these rhetors was to use sophists of the Classical Period, especially Gorgias, as stylistic models for emulation. Using Gorgias's oratory as a standard for rhetorical eloquence, Erling B. Holtsmark claims, was designed to induce a "renaissance of the Gorgianic ideals of prose" (Bryant 1968, pp. 52–53). Many such sophists were criticized by their contemporaries, particularly by those who saw the classical sophist Lysias as a better stylistic model and exemplar of a clearer, less ornate rhetoric. These individuals called themselves "Atticists" and were not exclusively from Athens or Rome but from the East as well. Dionysius of Halicarnassus, for example, was a proponent of a clear, direct style of rhetoric and a critic of Asiatic rhetoric. Second, it is apparent from a review of Eastern sophists that they demonstrated a range of expertise. Eastern sophists such as Xenocles of Adramyttium, Menippus of Stratonicea (Caria), and Molon of Rhodes were educators of prominent individuals—including, for example, Cicero. Sophists taught members of imperial families and were awarded not only chairs endowed by emperors and cities but also special privileges (Bowersock 1969, pp. 30–42). Third, sophists were not only eloquent orators but also theorists of such concepts as *stasis* theory and style (e.g., Hermagoras and Hermogenes), commentators on classical orators such as Demosthenes and Lysias (e.g., Zosimus of Gaza), and *presbyters* representing their cities in political matters (cf. Wooten 1987). Some Eastern sophists were Christian, such as Prohaeresius, the chairholder of Athens, and Procopius, who taught at the Christian sophistic school in Gaza.

A review of the reports of archaeological excavations of Ionian sites in modern Turkey also reveals a wealth of material evidence relating to the Second Sophistic. Centers that produced and attracted sophists—such as Ephesus, Pergamon, and Smyrna—reveal that their centers had municipal buildings that could function as sites for sophistic education and performance.[10] Strabo reveals that Pergamon and Ephesus had impressive libraries containing reading rooms and holding thousands of volumes in areas that were designed to minimize dampness to ensure preservation (Akurgal 1983, pp. 69–73, 159–61; cf. Strabo, *Geographia* 13.4.2, C624). Such sites invariably include gymnasia, odeions, theaters, and stoas. Many of these constructions were undertaken in the Hellenistic era but, importantly, were continually supported by Roman emperors and wealthy Roman patrons, as is evidenced by the Roman-sponsored reconstruction of Corinth (Papahatzis 1984). In addition to the physical remains at these sites, museums in Turkey also contain evidence of sophists and sophistic activities in the East that enrich our understanding of material rhetoric in the ancient world. For example, a statue of the sophist Flavius Damianus dating to about 200 CE was found in the east gymnasium of Ephesus and is now at the Archaeological Museum of Izmir (i.e., ancient Smyrna; Akurgal, Plate 52; see earlier Figure 2). Another statue of a sophist (also dated about 200 CE) was discovered in the imperial hall of the Vedius Gymnasium at Ephesus and is also on display at the Archaeological Museum of Izmir (Akurgal, Plate 53; see Figure 4). Of course, these items are only examples to make apparent the wealth of physical evidence awaiting discovery and analysis at many other archaeological sites and museums in Greece and Turkey by historians of rhetoric that will enrich our knowledge of the Second Sophistic in particular and the history of the discipline of rhetoric in general.

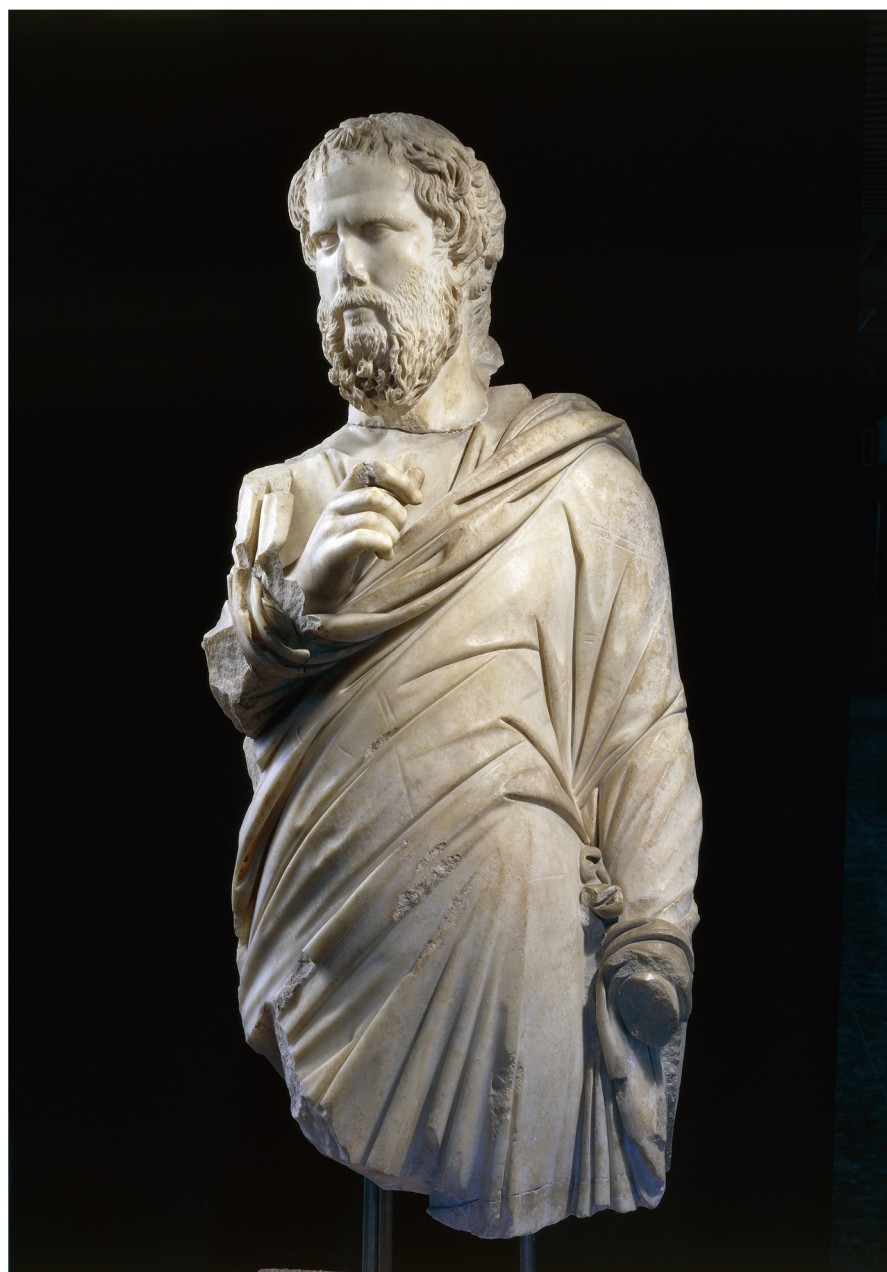

**Figure 4.** Incomplete marble statue of a sophist, 93–211 CE. Ephesus, Izmir Archaeological Museum. Art Resource, ART 420090. Used with permission.

### 9. Conclusions

A wealth of archaeological and epigraphical evidence unearthed over the last two centuries has revealed that the Second Sophistic not only occurred in and during the Roman Imperial Period but also that rhetorical teaching and performance by sophists was generally supported and even made possible by imperial, municipal, and individual patrons of the Roman Empire. It is not too much to claim that the pervasive and enduring influence of the Second Sophistic resulted because of Roman, and later Byzantine, support. For example, the findings from the Ionian cities of Anatolia—with Ephesus as the most vibrant example—show that archaeological evidence complements and reinforces literary *testimonia* with material rhetoric of physical artifacts and epigraphy. Ephesus, to cite the most stunning illustration, flowered during the height of the Roman Empire, and the remains of the auditorium, library, odeion, and great theater are evidence of a major site where sophistic education and performances would have taken place (Toksöz 1976, p. 12 *et passim*).

The extent of Roman governmental support and patronage provides a number of insights into the rhetorical features of the Second Sophistic.[11] Literary, archaeological, epigraphical, and historical evidence—in other words, material rhetoric—demonstrates that sophists were sponsored and often given privileges by influential Romans extending far beyond the granting of chairs of rhetoric. Public buildings such as those in Ephesus and Athens reveal athletic, aesthetic, and educational sites that supplemented politically functional civic structures. As the Roman Empire waned in the East and the Byzantine Empire rose to prominence, support for the Second Sophistic continued, despite the objections of some Church Fathers and the attempted invasions of barbarians in the East. In short, the evidence reveals that Roman imperial and individual patronage sustained the Second Sophistic as a movement in the Greek East and the Ionian sites of the Turquoise Coast. In short, the flourishing of rhetoric during the Second Sophistic was a direct consequence of Roman patronage at various levels.

The scholarship presented in this essay warrants several observations that make apparent research needs and opportunities for historians of rhetoric. As this essay has shown, research on the Second Sophistic in such eastern centers as Athens, Byzantium, and Antioch reveals the impact of the sophists on the history of rhetoric, law, politics, and education and culture, and therefore signals the need for continued work in and on these well-established centers of sophistic activity. More recent research efforts at such eastern sites as Rhodes, the Amphiareion of Oropos, Halicarnassus, Knidos, and Teos have provided further evidence of the breadth and influence of rhetoric, especially in terms of performance and education. This brief overview of material rhetoric in the Near East has also made it apparent that eastern centers—especially Ephesus and Smyrna—require further fieldwork and archival research. The benefits of continued research into these sites and lesser-known centers such as Pergamon, Abdera, and Tarsus offer the clear possibility of enriching not only our knowledge of the history of rhetoric but also related areas such as philosophy, religion, history, and cultural studies through a more thorough understanding of the context and climate of their occurrence in the Second Sophistic.

**Funding:** This research received no external funding.

**Institutional Review Board Statement:** Not applicable.

**Informed Consent Statement:** Not applicable.

**Data Availability Statement:** Not applicable.

**Conflicts of Interest:** The author declares no conflict of interest.

## Notes

1    It should be noted that the term "*kairos*" is not used in this essay in the conventional rhetorical sense of a "moment"or an "occasion" but rather as the social and mental "climate" of the period, much in the way that, for example, *kairos* is sometimes used in present-day Greece when discussing the "climate" in weather forecasts.

2    For a stimulating account of early efforts of Europeans to seek and better understand their Hellenic heritage, see David Constantine (2011), *In the Footsteps of the Gods: Travellers to Greece and the Quest for the Hellenic Ideal*.

3    Especially helpful in Cribiore's *The School of Libanius in Late Antique Antioch* is chapter two, "Schools and Sophists in the Roman East", and chapter three, "The Network."

4    For excellent examples applying reception theory to the study of rhetoric, see Papaioannou et al., eds. 2022, *Brill's Companion to the Reception of Ancient Rhetoric* (Papaioannou et al. 2022).

5    The democracy of Athens made debate a common civic activity, one that sophists taught with expertise.

6    In his 2005 work, *Rhetoric in Antiquity*, Pernot refers to the Second Sophistic of the Imperial Age as being "sufficiently numerous and important to form a movement and to serve as models" (p. 187). In his 2017 essay, Pernot reaffirms his view that the Second Sophistic was a "literary and social movement", albeit "not an organized social movement" (pp. 253–54). These two statements by Pernot, however, are not incompatible. Social movements do not necessarily need to be an organized collectivity as a necessary criterion, nor does their impact diminish because of the absence of a formal, institutionally sanctioned entity. Rather, as Pernot points out, the sophists of the Second Sophistic do have shared features and a shared hierarchy of values and standards (Pernot

2005, pp. 186–94; 2017, p. 254). In this case, the models of the First Sophistic of the Classical Period served as a source of shared values and standards for the sophists of the Second Sophistic.

7     A review of the features that constitute a social movement in *Persuasion and Social Movements* by Charles J. Stewart, Craig Allen Smith, and Robert E. Denton, Jr. makes it readily apparent that the Bolshevik revolution of Russia, the civil-rights movement in the USA, and the Polish freedom movement that achieved independence from the USSR are all clear examples of social movements that are rhetorical in nature (Stewart et al. 2012).

8     Examples of *progymnasmata* or sophistic *téchnai* are offered in George A. Kennedy's *Progymnasmata: Greek Textbooks of Prose Composition*. The sophists whose works appear in this collection include Theon, Hermogenes, Aphthonius, and Nicolaus.

9     The summary of the traits and achievements of Eastern sophists presented in this paragraph draws heavily from the work of Bryant et al. in *Ancient Greek and Roman Rhetoricians: A Bibliographical Dictionary*. Readers may also wish to consult Murphy in Project Rhetor, *A Bibliographical Dictionary of Greek and Roman Rhetoricians: A Preliminary Survey*.

10    Pernot regards Smyrna, Ephesus, and Athens as the "principal centers" of the Second Sophistic (Pernot 2017, p. 254).

11    A more detailed discussion of the observations made here, with particular emphasis placed on Athens, can be found in Enos, *Roman Rhetoric: Revolution and the Greek Influence* (Enos 2008, pp. 138–51).

## References
### 1. Classical Sources

Aristophanes. *Clouds (Nubes)*.
Athenaeus. *Deipnosophistae*.
Cicero. *Brutus*.
Eunapius. *Vitae sophistarum*.
Lucian, *My Dream* (*Somnium*).
Lucian, *The Professor of Rhetoric* (*Rhetorum praeceptor*).
Lucian, *The Pseudo-Sophist or Solecist* (*Pseudosophostes e Soloikisthes*).
Pausanias. *Description of Greece*.
Philostratus. *Vitae sophistarum*.
Quintilian. *Institutio oratoria*.
Strabo. *Geographia*.
Suidas. *Suda* (Byzantine Lexicography).

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
