# Peer review of "A Site-Perspective on the Second Sophistic of the near East and Its Impact on the History of Rhetoric: An Overview"

_humanities, doi:10.3390/h11060154_

Round 1
Reviewer 1 Report
Review of “A Site-Perspective on the Second Sophistic of the Near East and Its Impact on the History of Rhetoric: An Overview”
Historians of rhetoric ought to conduct plentifully more research, including site-based, on ancient cities other than Athens and Rome. On that point, the author and I agree. The paper does not fill me with confidence, however, that the author is well-positioned to issue such a call. Their trustworthiness is lacking. Many aspects of this paper compel me to reach that conclusion, but I will limit myself to three: 1) ironically, the piece uncritically features Western triumphalism and paternalism; 2) the author strings together secondary sources and does not directly engage anything primary (whether textual, visual, or material); 3) the author does not seem to know the nature of several concepts used in the piece.
First, this piece tilts toward Western triumphalism and paternalism, and sometimes not subtly. It shows in places like on p. 3, where the author writes of “Efforts by Westerners to brave the perils of the Ottoman Empire…”. It shows in “the Greek miracle.” Any time anyone uses the term “the Greek miracle,” they need to explain that notion and the (rightful) controversy around it. When someone uses the term and does not explain its controversial nature, then I read the term as conveying an attitude that ancient Greece was the start and origin of everything culturally valuable, which is simply not true. It shows in how the author continually credits Roman imperium with the existence and sustenance of “the second sophistic” without reflecting on that judgment. Given that the author cites Jarratt’s Chain of Gold, I would expect careful qualification of such a claim.
Second, the author strings together secondary sources rather than using them complement or supplement their own argument. Purported overviews of previous scholarship read like minimally annotated bibliographies. (There are several highly pertinent books from recent years that the author does not seem to know about from philosophers and classicists, such as Christopher Moore’s book Calling Philosophers Names, which is largely about “sophos”-based terms, and Laura Viidebaum’s Creating the Ancient Rhetorical Tradition.) That is one issue. A related problem is that the author does not engage anything primary, and especially not in a way that seems to have resulted from their own site-based work. If one is making a case for site-work, then a reader would expect the author to have done some and to be able to draw upon the work they conducted. There are rhetoricians (and other scholars) who have done and are doing site-based work to enhance and improve our understanding of ancient rhetorical cultures, and it reads very differently from this piece. There are rhetoricians (and other scholars) who have done and are doing work on epigraphy and statues, and it, too, reads very differently from this piece.
Third, the author’s use of several keywords from the study of rhetoric suggests they do not know what the terms literally mean or how they were used or are being used. Sometimes the issue is one of anachronism. For instance, to translate “technÄ“” (and the author never indicates the eta) as “discipline” cannot be substantiated without a good description of what constitutes a “discipline” in the 5th century BCE. The word “kairos” is misused. “Metics” does not mean “foreign educators.” “Social movement” is misused. (Kerford, who’s not in the bibliography, did write a book about “the sophistic movement” of the 5th century BCE, but he means something vastly different than a contemporary social movement. The literature on social movements is vast, but the author would find nothing in that literature to indicate disconnected groups of “sophists” with no collective identity or agenda constituted a social movement.) Some of the titles of ancient Greek texts the author cites are given in English and then in Latin, which is curious. Perhaps the biggest issue is that I’m not at all confident the author knows what “the second sophistic” means, where that name came from, what it was doing pragmatically, culturally, and ideologically, and the current state of the scholarship on what it means. It was p. 13 that most gave me pause and trouble. For example, saying that a political entity like a state gave “support for the Second Sophistic” does not make sense. The second sophistic was not a name that people called themselves.
I found the chronology of the whole piece hard to follow. It’s ducking in and out of centuries and decades as if “sophist” means the same thing in every place and time. The historical work is sloppy and derivative.
I really wanted to be able to support this piece, but it goes about what it’s ostensibly trying to do in a strange and sometimes troubling way.
Author Response
This revision is based upon the suggestions of the editor and the two readers. I wish to thank them all for their thorough and careful reading of my essay and the thoughtful recommendations that they have provided. I believe that their insights have improved the quality of my work and I wish here to extend my thanks and appreciation. For the sake of clarity I will list the changes, not in any hierarchy, in this attached revision.
- I have condensed the abstract and revised it not only to make clear the nature of the work, but also the implications of my findings and recommendations.
- I have sought to make clear that this essay is, especially in its initial stages, an overview of established and, in some cases, tacitly accepted views and methods that need to be reassessed. To that end, the essay does (at times) appear as an annotated review but that approach is necessary in order to help the reader to understand past work, not only for the contributions but for the needs that remain. Specifically, that while understanding the traits of individual sophists—both those of the Classical Period and of the Second Sophist—their remains a need to engage in field and archival research that seeks new primary evidence not available in established literary testimonia but in situ artifacts (that are termed here material rhetoric) and contributes to contextualizing and situating contributions.
- I have modified and revised the discussion of the Second Sophistic in terms of social/cultural movement but, after over twenty years of teaching undergraduate and graduate courses in persuasion and social movements, I feel confident that my observations, as well as my comments on the work of Pernot and others, is fair (generally supportive) and accurate.
- I have been asked to elaborate on the contributions of Susan Jarratt, specifically her Chain of Gold, and I have done that explicitly and where appropriate. I also have incorporated the recommendations to review the insights of the two recently published works of Christopher Morris (Calling Philosophers Names: On the Origins of a Discipline, 2020) and Laura Viidebaum (Creating the Ancient Rhetorical Tradition, 2021). I have since purchased both works and found both of these new works very helpful in establishing and reinforcing my claims. Their respective observations are included in this revision with respect to the term sophos and sophists of the Classical Period. I extend my appreciation for reading these suggestions. In the same spirit, I have also provided more detail about the term metic.
- I have been asked to include a synoptic example of my own work that will provide an illustration of the benefits of the site-perspective procedures that I call for in this overview essay. To that end, I have selected one such project that I did in Greece and done through the approval and cooperation of The American School of Classical Studies at Athens and (what is now termed) The Hellenic Republic Ministry of Culture and Sports. NB: I do have other projects that involve working with archaeologists from Turkey but not in situ because of medical and political conditions, as well as ongoing research but that work is still under development.
- Because the Second Sophistic is referential in many respects to the earlier Classical Period, it is difficult to offer a simple linear timeline. I am sympathetic to this condition but appreciating the discrete points made in individual rubrics requires a degree of flexibility and (I am sure) tolerance. That said, I hope that the closing comments of my conclusion will synthesize and clarify these points.
- In general terms, I have revised the essay for overall organization and clarity. For the most part this may not initially appear as major changes but I believe that the hours devoted to editing and elaborating individual items will result in a more coherent, unified effect to the work.
Again, my thanks and appreciation for the above recommendations that, I believe, have improved the essay.
Reviewer 2 Report
The article is exceptional and achieves its objectives in an admirable manner.
The only aspect I did not really respond well to is the abstract, which seems to be verbose and explains what it intends to achieve instead of just doing it! (I do not like abstracts that set out what the article intends to discussion; rather, I would just like to see an summary of the content and a brief explanation of what it achieves in terms of its scholarship.) The abstract could be shortened and tightened as well as unnecessary verbiage removed.
I am not a fan of the first-person, royal 'we' style, but I realize that this style is widely employed in academic writing in our age.
I must stress, however, that the content is superb and readers will benefit enormously from the discussion on the Second Sophistic in the Near East. Most treatments deal with the Second Sophistic in other parts of the Empire, so much is added to the overall background and context of the Second Sophistic through the discussion that takes place in this article.
Author Response

(The authors gave the same response as above.)

Reviewer 3 Report
A very useful and encyclopedic treatment of the Second Sophistic, with equally salient framing of the importance of site specific research. Any student or scholar seeking to investigate this under explored domain will be well served by this contribution.
The exigence of the essay is clear: While many disciplines have invoked a "Third Sophistic" of rhetoric, such invocations often offer little account of the Second Sophistic or its Gorgianic forebears. Too often, debates about "sophistry" fall into a "Plato vs the Sophists" dynamic, ignoring both the sophistic aspects of Plato's own rhetorical practices and homogenizing "The Sophists" into a dialectical other to Plato, whose own articulations of sophistic practice shift within a single dialogue ( Phaedrus), remain enigmatic ( Gorgias) or even links pure recitation to the divine (Ion). This essay moves beyond such a dynamic and explores the rich and diverse history of the Second Sophistic itself through a site specific methodology that begins to integrate the wide ranging scholarship on sophistic rhetorics, with accounts of the secondary literature, helpful guides to sources, and a cautionary account of the diversity of approaches and schools that elude any simple account. I found the account of the Christian schools and approaches in the Second Sophistic to be particularly intriguing, and its treatment of the "theatrical" and competitive aspect of the rhetorical games helpfully links it to a "skaldic" tradition of rhetoric hinted at by scholars of the Upanishads (e.g. Nagler) as the creative genesis for spiritual and philosohical texts. The exigence, scope and diversity of the essay's account offers an eponymous "overview" for any scholar in rhetorical studies, philosophy, comparative literature, religious studies, history, or literary criticism who seeks to go beyond the debates and take a deeper dive into the Second Sophistic before, or even as, they invoke the Third.
The research, besides the site analysis and its emphasis on the rhetorical games, is not path breaking but integrative. Hence its strength - the organization and articulation of the overview - can also be a weakness, as the essay is strongest on integrating its own site analysis into the larger literature. Still, this integration and emphasis on rhetorical games is a boon, and my rationale for urging publication. This is an essay I will want to share with graduate students, advanced undergraduates and colleagues whenever they have questions about or interests in the sophists.